# New sulphonamide pyrolidine carboxamide derivatives: Synthesis, molecular docking, antiplasmodial and antioxidant activities

**Efeturi A. Onoabedje[1,2]\***, **Akachukwu Ibezim**[3]**\***, **Uchechukwu C. Okoro**[1], **Sanjay Batra**[2]

**1** Department of Pure & Industrial Chemistry, Faculty of Physical Sciences, University of Nigeria, Nsukka, Enugu State, Nigeria, **2** Division of Medicinal & Process Chemistry, Central Drug Research Institute, Lucknow, UP, India, **3** Department of Pharmaceutical and Medicinal Chemistry, Faculty of Pharmaceutical Sciences, University of Nigeria, Nsukka, Enugu State, Nigeria

\* efeturi.onoabedje@unn.edu.ng (EAO); akachukwu.ibezim@unn.edu.ng (AI)

**Data Availability Statement:** All relevant data are within the manuscript and its Supporting Information files.

## Abstract

Carboxamides bearing sulphonamide functionality have been shown to exhibit significant lethal effect on *Plasmodium falciparum*, the causative agent of human malaria. Here we report the synthesis of thirty-two new drug-like sulphonamide pyrolidine carboxamide derivatives and their antiplasmodial and antioxidant capabilities. In addition, molecular docking was used to check their binding affinities for homology modelled *P. falciparum* N-myristoyl-transferase, a confirmed drug target in the pathogen. Results revealed that sixteen new derivatives killed the parasite at single-digit micromolar concentration ($IC_{50}$ = 2.40–8.30 µM) and compounds **10b**, **10c**, **10d**, **10j** and **10o** scavenged DPPH radicals at $IC_{50}$s (6.48, 8.49, 3.02, 6.44 and 4.32 µg/mL respectively) comparable with 1.06 µg/mL for ascorbic acid. Compound **10o** emerged as the most active of the derivatives to bind to the PfNMT with theoretical inhibition constant ($K_i$ = 0.09 µM) comparable to the reference ligand pyrazole–sulphonamide ($K_i$ = 0.01 µM). This study identifies compound **10o**, and this series in general, as potential antimalarial candidate with antioxidant activity which requires further attention to optimise activity.

## Introduction

Malaria is a human parasitic disease that is caused by some species of *Plasmodium* in which *Plasmodium falciparum* is the deadliest [1, 2]. In spite of extensive measures to combat the disease [3, 4], 214 million new cases and more than 400,000 deaths were reported in 2018 [5]. The emergence of drug-resistance strains of *P. falciparum*, which are no longer susceptible to even frontline drugs such as artemisinin, and cross-resistance, where resistance to one drug confers resistance to other chemically similar drugs or those that share mode of action, are blamed for the persistent devastation [6] and raises the dire need to discover new drugs to check the rate of malaria morbidity as well as mortality [7, 8]. Within red blood cells (RBCs), *P. falciparum* transforms from ring stage to trophozoite, then to schizont and finally to merozoites that egress and invade RBCs. The merozoite invasion of RBCs is driven by molecular motor

**Funding:** EAO received funding from CSIR-TWAS in the form of postdoctoral stay in a lab. in India. AI received funding from AGNES through Junior Researchers Grant (JRG). Computational resources were provided by Dr. F. Ntie-Kang.

**Competing interests:** The authors have declared that no competing interests exist.

complex which has been assembled by inner membrane complex (IMC). The human malaria parasite requires protein myristoylation for life stage progression. N-myristoltransferase (NMT) catalyses myristoylation process i.e. the co- and post-translational transfer of myristoyl group from the cofactor to the N-terminal glycine protein substrate. NMT inhibition induces loss of IMC activities and hence parasite viability [9]. Also, some proteins which are essential for parasite to attach to new RBCs are found missing when NMT is inhibited [10, 11]. The foregoing provides strong evidence that NMT inhibited *P. falciparum* is non-infectious and therefore NMT is a good drug target for antimalaria drug search. Likewise, malaria infection increases the amount of free radicals because the activities of *Plasmodium* stimulates the generation of free radicals which are essential in malaria physiopathogenesis [12, 13]. Similarly, Clark and coworkers [14] observed that antioxidant enzymes highly decrease in malaria patients. So, it is advantageous for compounds to possess both antiplasmodial and antioxidant properties and in fact researchers presently screen for both activities in a putative antimalarial agent.

Sulphonamides are group of sulfa drugs used in the treatment of infections since 1930s. They have been reported to possess anti-virial, antihypertensive, anticancer, antiprotozoal, antimicrobial, antioxidant and carbonic anhydrase inhibitory properties [15]. Similarly, numerous natural and synthetic carboxamides have demonstrated notable degree of inhibitory potency against *P. falciparum*. For instance, leukinnostatin A, efrapeptin, zeryamicin, and antinmoebin are few examples of linear carboxamides are found to be lethal to the pathogen at low micromolar concentration [16–20]. While sulphonamides exert their antimalarial activity through disruption of the folate biosynthesis [21], carboxamides have been reported to inhibit haemoglobin degradation by plasmepsin and cysteine proteases in acidic food vacuole and both pathways are vital for the parasite survival [22, 23]. Consequently, carboxamides bearing sulphonamide functionality are hypothesised to show improved antimalarial effect through enhanced proteolytic stability and hydrogen-bonding and our past studies as well as other research reports have confirm that to be true [24, 25]. Therefore, search for antiplasmodial agents in sulphonamide-carboxamide hybrid is highly desirable.

In this present work, we continued our search for new antimalarial lead that possesses antioxidant effect in the sulphonamide-carboxamide series [26, 27] and in addition tested their capability to bind to *P. falciparum* NMT (PfNMT); a validated antimalarial drug target.

## Material and methods

### General information

Commercially available chemicals were purchased from Sigma-Adrich and Spectrochem Pvt Ltd (India) and used without further purification. Unless otherwise stated all reactions were performed in non-dry glassware under an air atmosphere and were monitored by analytical thin layer chromatography (TLC). TLC was performed on pre-coated silica gel plates. After elution, plate was visualized under UV illumination at 254 nm for UV active materials. The melting points were recorded on a hot stage apparatus and are uncorrected. IR spectra were recorded using Agilent Cary 630 FTIR spectrophotometer. $^1$H NMR and $^{13}$C NMR spectra were recorded on Bruker Av III HD 400 MHz NMR spectrometers with DMSO-$d^6$ as solvent, using TMS as an internal standard (chemical shifts in δ). Peak multiplicities of $^1$H-NMR signals were designated as s (singlet), brs (broad singlet), d (doublet), dd (doublet of doublet), t (triplet), q (quartet), m (multiplet) etc. Coupling constants (*J*) are in Hz. The ESI-MS were recorded on Triple Quadrupole Mass spectrometer. Column chromatography was performed using silica gel (100–200 mesh). Analytical grade solvents for the column chromatography were used as received.

## General procedures for the synthesis

To 2-(4-methylphenylsulfonamido)propanoic acid (1.0 equiv.), EDC.HCl (1.2 equiv) and HOBT (1.2 equiv) was added DCM (5.0 mL) and DIPEA (2.0 equiv.) and the reaction mixture stirred for 20 min at room temperature. Thereafter, 2-amino-*N*-substitutedphenylpropanamide (1.0 equiv.) was added and the entire reaction was allowed to stir at room temperature overnight. The reaction mixture was diluted with DCM (20mL), and subsequently washed with 5% bicarbonate, 1.0 M HCl and brine. The organic layer was dried over $Na_2SO_4$, filtered and evaporated under vacuum to provide the crude product, which was purified by column chromatography (10–70% EA/Hexane) to afford the desire product.

**N-(1-oxo-1-(phenylamino)propan-2-yl)-1-tosylpyrrolidine-2-carboxamide (9a).** Yield: 68%; a white solid, mp 122–124°C; $R_f$ = 0.48 (Hexane: EtOAc, 3:7, v/v). IR (CHCl₃) $\nu_{max}$: 1216, 1341 (S = O), 1677 (C = O), 3369 (N-H) cm⁻¹.¹H NMR (400 MHz, DMSO-d₆): $\delta$ = 1.35 (d, *J* = 7.1 Hz, 3H), 1.47–1.52 (m, 1H), 1.69–183 (m, 3H), 2.42 (s, 3H), 3.12–3.18(m, 1H), 3.37–3.45(m,1H), 4.14 (dd, $J^1$ = 8.2 Hz; $J^2$ = 3.57 Hz, 1H), 4.46 (p, *J* = 7.1 Hz, 1H), 7.06 (t, *J* = 7.4 Hz, 1H), 7.32 (t, *J* = 7.7 Hz, 2H), 7.45(d, *J* = 8.1 Hz, 2H), 7.62 (d, *J* = 7.8 Hz, 2H), 7.77 (d, *J* = 8.2 Hz, 2H), 8.29 (d, *J* = 7.4 Hz, 1H), 9.87 (s, 1H). ¹³C NMR (100 MHz, DMSO-d₆): 18.7, 21.5, 24.6, 31.1, 49.4, 49.6, 61.5, 119.7, 123.9, 127.9, 129.2, 130.3, 134.3, 139.3, 144.1, 171.4, 171.5 (C = O). MS (ESI+): *m/z* = 416.2. ESI-HR-MS calculated for $C_{21}H_{25}N_3O_4S$ (M⁺+H): 416.1644, found: 416.1637.

**N-(1-oxo-1-(p-tolylamino)propan-2-yl)-1-tosylpyrrolidine-2-carboxamide (9b).** Yield: 65% Yield: 67%; a white solid, mp 95–97°C; $R_f$ = 0.47 (Hexane: EtOAc, 3:7, v/v). IR (CHCl₃) $\nu_{max}$: 1217, 1341 (S = O), 1670 (C = O), 3365 (N-H) cm⁻¹.¹H NMR (400 MHz, DMSO-d₆): $\delta$ = 1.33(d, *J* = 7.1 Hz, 3H), 1.46–1.51 (m, 1H), 1.68–182 (m, 3H), 2.25 (s, 3H),2.41 (s, 3H), 3.12–3.17 (m, 1H), 3.39–3.44(m, 1H), 4.12 (dd, $J^1$ = 8.2 Hz; $J^2$ = 3.7 Hz, 1H), 4.44 (p, *J* = 7.2 Hz, 1H), 7.11 (d, *J* = 8.3 Hz, 2H), 7.43–7.50 (m, 4H), 7.77 (d, *J* = 8.2 Hz, 2H), 8.26 (d, *J* = 7.5 Hz, 1H), 9.77 (s, 1H). ¹³C NMR (100 MHz, DMSO-d₆): 18.7, 20.9, 21.5, 24.6, 31.1, 49.3, 49.6, 61.6, 119.7, 127.9,129.6, 130.3, 132.8, 134.3, 136.8, 144.1, 171.1, 171.4 (C = O). MS (ESI+): *m/z* = 430.4. ESI-HR-MS calculated for $C_{22}H_{27}N_3O_4S$ (M⁺+H): 430.1801, found: 430.1798.

**N-(1-(4-methoxyphenylamino)-1-oxopropan-2-yl)-1-tosylpyrrolidine-2-carboxamide (9c).** Yield: 81%; a white solid, mp 79–81°C; $R_f$ = 0.54 (Hexane: EtOAc, 3:7, v/v). IR (CHCl₃) $\nu_{max}$: 1217, 1342 (S = O), 1674 (C = O), 3373 (N-H) cm⁻¹.¹H NMR (400 MHz, DMSO-d₆): $\delta$ = 1.33(d, *J* = 5.9 Hz, 3H), 1.48 (s, 1H), 1.72–1.78 (m, 3H), 2.41 (s, 3H), 3.15 (d, *J* = 6.6 Hz, 1H), 3.42(s, 1H), 3.72 (s, 3H), 4.12 (t, *J* = 4.1 Hz, 1H), 4.45 (t, *J* = 6.3 Hz, 1H), 6.89 (d, *J* = 7.1 Hz, 2H), 7.45 (d, *J* = 7.7 Hz, 2H), 7.52 (d, *J* = 7.7 Hz, 2H), 7.77 (d, *J* = 6.9 Hz, 2H), 8.26 (d, *J* = 6.5 Hz, 1H), 9.72 (s, 1H). ¹³C NMR (100 MHz, DMSO-d₆): 18.8, 21.5, 24.6, 31.1, 49.2, 49.6, 55.6, 61.6, 114.4, 121.2, 127.9, 130.3, 132.4, 134.3, 144.1, 155.8, 170.8, 171.4 (C = O). MS (ESI+): *m/z* = 446.3. ESI-HR-MS calculated for $C_{22}H_{27}N_3O_4S$ (M⁺+H): 446.1750, found: 446.1746.

**N-(1-(4-isopropylphenylamino)-1-oxopropan-2-yl)-1-tosylpyrrolidine-2-carboxamide (9d).** Yield: 74%; a white solid, mp 121–123°C; $R_f$ = 0.44 (Hexane: EtOAc, 3:7, v/v). IR (CHCl₃) $\nu_{max}$: 1246, 1339 (S = O), 1667 (C = O), 3359 (N-H) cm⁻¹.¹H NMR (400 MHz, DMSO-d₆): $\delta$ = 1.10(d, *J* = 6.9 Hz, 6H), 1.7(d, *J* = 6.9 Hz, 3H), 1.39–1.45 (m, 1H), 1.62–1.75 (m, 3H), 2.5 (s, 3H), 2.7–2.80(m, 1H), 3.04–3.10(m, 1H), 3.30–3.37(m, 1H), 4.06 (dd, $J^1$ = 8.2 Hz; $J^2$ = 3.8 Hz, 1H), 4.37 (p, *J* = 7.3 Hz, 1H), 7.11 (d, *J* = 8.6 Hz, 2H), 7.38 (d, *J* = 8.0 Hz, 2H), 7.44 (d, *J* = 8.6 Hz, 2H), 7.70 (d, *J* = 8.2 Hz, 2H), 8.19 (d, *J* = 7.5 Hz, 1H), 9.71 (s, 1H). ¹³C NMR (100 MHz, DMSO-d₆): 18.7, 21.5, 24.4, 24.6, 31.1, 33.2, 49.3, 49.6, 61.6, 119.8, 126.9, 127.9, 130.3, 134.4, 137.0, 144.0, 144.1, 171.1, 171.5 (C = O). ESI-HR-MS calculated for $C_{24}H_{31}N_3O_4S$ (M⁺+H): 458.2114, found: 458.2110.

**N-(1-(4-chlorophenylamino)-1-oxopropan-2-yl)-1-tosylpyrrolidine-2-carboxamide (9e).** Yield: 71%; a white solid, mp 90–92°C; $R_f$ = 0.41 (Hexane: EtOAc, 3:7, v/v). IR (CHCl₃)

$v_{max}$: 1158, 1339 (S = O), 1667 (C = O), 3355 (N-H) cm$^{-1}$.$^{1}$H NMR (400 MHz, DMSO-d$_6$): $\delta$ = 1.35 (d, $J$ = 7.1 Hz, 3H), 1.47–1.52 (m, 1H), 1.69–182 (m, 3H), 2.42 (s, 3H), 3.12–3.18(m, 1H), 3.38–3.44(m, 1H), 4.14 (dd, $J^1$ = 8.2 Hz; $J^2$ = 3.74 Hz, 1H), 4.44 (p, $J$ = 7.2 Hz, 1H), 7.37–7.40 (m, 2H), 7.45 (D, $J$ = 8.2 Hz, 2H), 7.62–7.66 (m, 2H), 7.77 (d, $J$ = 8.2 Hz, 2H), 8.31 (d, $J$ = 7.3 Hz, 1H), 10.03 (s, 1H). $^{13}$C NMR (100 MHz, DMSO-d$_6$): 18.5, 21.5, 24.6, 31.3, 49.4, 49.6, 61.5, 121.2, 127.4, 127.7, 129.1, 130.3, 134.4, 138.2, 144.1, 177.5 (C = O). ESI-HR-MS calculated for C$_{21}$H$_{24}$ClN$_3$O$_4$S (M$^+$+H): 450.1254, found: 450.1250.

**N-(1-(4-fluorophenylamino)-1-oxopropan-2-yl)-1-tosylpyrrolidine-2-carboxamide (9f).** Yield: 68%; a white solid, mp 110–112˚C; R$_f$ = 0.56 (Hexane: EtOAc, 3:7, v/v). IR (CHCl$_3$) $v_{max}$: 1214, 1339 (S = O), 1666 (C = O), 3357 (N-H) cm$^{-1}$.$^{1}$H NMR (400 MHz, DMSO-d$_6$): $\delta$ = 1.34(d, $J$ = 6.7 Hz, 3H), 1.47 (s, 1H), 1.72–178 (m, 3H), 2.42 (s, 3H), 3.12–3.8 (m, 1H), 3.37–3.45 (m, 1H), 4.14 (dd, $J^1$ = 8.2 Hz; $J^2$ = 3.57 Hz, 1H), 4.46 (q, $J$ = 7.1 Hz, 1H), 7.06 (t, $J$ = 7.4 Hz, 1H), 7.32 (t, $J$ = 7.7 Hz, 2H), 7.45 (d, $J$ = 8.1 Hz, 2H), 7.62 (d, $J$ = 7.8 Hz, 2H), 7.77 (d, $J$ = 8.2 Hz, 2H), 8.229 (d, $J$ = 7.4 Hz, 1H), 9.87 (s, 1H). $^{13}$C NMR (100 MHz, DMSO-d$_6$): 18.7, 21.5, 24.6, 31.1, 49.4, 49.6, 61.5, 119.7, 123.9, 127.9, 129.2, 130.3, 134.3, 139.3, 144.1, 171.4, 171.5 (C = O). MS (ESI+): $m/z$ = 416.2. ESI-HR-MS calculated for C$_{21}$H$_{25}$N$_3$O$_4$S (M$^+$+H): 416.1644, found: 416.1637.

**N-(1-(4-fluorophenylamino)-1-oxopropan-2-yl)-1-tosylpyrrolidine-2-carboxamide (9g).** Yield: 54%; a white solid, mp 81–83˚C; R$_f$ = 0.65 (Hexane: EtOAc, 3:7, v/v). IR (CHCl$_3$) $v_{max}$: 1217, 1335 (S = O), 1679 (C = O), 3361 (N-H) cm$^{-1}$.$^{1}$H NMR (400 MHz, DMSO-d$_6$): $\delta$ = 1.37(d, $J$ = 7.0 Hz, 3H), 1.47–1.52 (m, 1H), 1.70–1.83 (m, 3H), 2.42(s, 3H), 3.13–3.18(m, 1H), 3.38–3.44 (m, 1H), 4.13–4.16 (m, 1H), 4.15 (dd, $J^1$ = 8.1 Hz; $J^2$ = 3.5 Hz, 1H), 4.41–4.48 (m, 1H), 7.44 (t, $J$ = 8.3 Hz, 3H), 7.75 (t, $J$ = 8.0 Hz, 1H), 7.78 (t, $J$ = 11.1 Hz, 3H),8.11 (s, 1H), 8.33 (d, $J$ = 7.1 Hz, 1H), 10.25 (s, 1H). $^{13}$C NMR (100 MHz, DMSO-d$_6$): 18.4, 21.5, 22.5, 24.6, 31.1, 31.4, 49.6, 61.4, 115.7, 120.2, 123.2, 127.7, 127.9, 129.8, 130.1, 130.3, 130.5, 134.4, 140.1, 144.1, 171.6, 172.0 (C = O). MS (ESI+): $m/z$ = 484.4. ESI-HR-MS calculated for C$_{21}$H$_{25}$N$_3$O$_4$S (M$^+$+H): 484.1518, found: 484.1513.

**N-(1-(4-chlorobenzylamino)-1-oxopropan-2-yl)-1-tosylpyrrolidine-2-carboxamide (9h).** Yield: 54%; a white solid, mp 68–70˚C; R$_f$ = 0.42 (Hexane: EtOAc, 3:7, v/v). IR (CHCl$_3$) $v_{max}$: 1159, 1216, 1344 (S = O), 1671 (C = O), 3399 (N-H) cm$^{-1}$.$^{1}$H NMR (400 MHz, DMSO-d$_6$): $\delta$ = 1.27 (d, $J$ = 7.0 Hz, 3H), 1.45–1.48 (m, 1H), 1.67–1.78 (m, 4H), 2.41 (s, 3H), 3.11–3.16 (m, 1H), 4.07–4.09 (m, 1H), 4.23–4.33 (m, 3H), 7.27 (d, $J$ = 8.2 Hz, 2H), 7.37 (d, $J$ = 8.2 Hz, 2H), 7.44 (d, $J$ = 8.0 Hz, 2H), 7.74 (d, $J$ = 8.1 Hz, 2H), 8.16 (d, $J$ = 7.32 Hz, 1H), 8.36 (t, $J$ = 5.5 Hz, 1H). $^{13}$C NMR (100 MHz, DMSO-d$_6$): 18.6, 21.5, 24.6, 31.1, 41.8, 48.8, 49.6, 61.6, 127.9, 128.7, 129.4, 130.3, 131.8, 134.3, 138.8, 144.1, 171.4, 172.5 (C = O). MS (ESI+): $m/z$ = 464.4. ESI-HR-MS calculated for C$_{22}$H$_{26}$ClN$_3$O$_4$S (M$^+$+H): 464.1411, found: 464.1408.

**N-(1-(3,4-dichlorophenylamino)-1-oxopropan-2-yl)-1-tosylpyrrolidine-2-carboxamide (9i).** Yield: 61%; a white solid, mp 98–100˚C; R$_f$ = 0.50 (Hexane: EtOAc, 3:7, v/v). IR (CHCl$_3$) $v_{max}$: 1341, 1383 (S = O), 1680 (C = O), 3361 (N-H) cm$^{-1}$.$^{1}$H NMR (400 MHz, DMSO-d$_6$): $\delta$ = 1.28(d, $J$ = 7.1 Hz, 3H), 1.40–1.45 (m, 1H), 1.63–1.75 (m, 3H), 2.35 (s, 3H), 3.05–3.11 (m, 1H), 3.31–3.35 (m, 1H), 4.07 (dd, $J^1$ = 8.1 Hz; $J^2$ = 3.7 Hz, 1H), 4.34 (p, $J$ = 7.0 Hz, 1H), 7.38 (d, $J$ = 8.1 Hz, 2H), 7.44(dd, $J^1$ = 8.9 Hz; $J^2$ = 2.4 Hz, 1H), 7.52 (d, $J$ = 8.8 Hz, 1H), 7.70 (d, $J$ = 8.3 Hz, 2H), 7.93 (d, $J$ = 2.4 Hz, 1H), 8.26 (d, $J$ = 7.2 Hz, 1H), 10.13 (s, 1H). $^{13}$C NMR (100 MHz, DMSO-d$_6$): 18.3, 21.5, 24.6, 31.1, 49.6, 61.4, 119.7, 120.9, 125.4, 127.9, 130.3, 131.5, 134.4, 139.4, 144.1, 171.6, 172.0 (C = O). ESI-HR-MS calculated for C$_{21}$H$_{23}$Cl$_2$N$_3$O$_4$S (M$^+$+H): 484.0865, found: 484.0869.

**N-(1-(3-chloro-4-fluorophenylamino)-1-oxopropan-2-yl)-1-tosylpyrrolidine-2-carbox-amide (9j).** Yield: 54%; a white solid, mp 153–155˚C; R$_f$ = 0.53 (Hexane: EtOAc, 3:7, v/v). IR (CHCl$_3$) $v_{max}$: 1158, 1218, 1394 (S = O), 1670 (C = O), 3358 (N-H) cm$^{-1}$.$^{1}$H NMR (400 MHz,

DMSO-d$_6$): $\delta$ = 1.35 (d, $J$ = 7.1 Hz, 3H), 1.47–1.52 (m, 1H), 1.72–1.82 (m, 3H), 2.42 (s, 3H), 3.12–3.18 (m, 1H), 3.38–3.45 (m, 1H), 4.14 (dd, $J^1$ = 8.1 Hz; $J^2$ = 3.8 Hz, 1H), 4.42 (p, $J$ = 7.0 Hz, 1H), 7.37–7.48 (m, 4H), 7.77 (d, $J$ = 8.3 Hz, 2H), 7.94 15 (dd, $J^1$ = 6.9 Hz; $J^2$ = 2.6 Hz, 1H), 8.33(d, $J$ = 7.3 Hz, 1H), 10.13 (s, 1H). $^{13}$C NMR (100 MHz, DMSO-d$_6$): 18.4, 21.5, 24.6, 31.1, 49.5, 49.6, 61.4, 119.5, 119.7, 119.9, 120.0, 121.0, 127.9, 130.3, 134.4, 136.6, 144.1, 152.4, 171.6, 171.7 (C = O). MS (ESI+): $m/z$ = 468.4. ESI-HR-MS calculated for $C_{21}H_{23}ClFN_3O_4S$ (M$^+$+H): 468.1160, found: 468.1164.

**$N$-(1-(3,4-dimethoxyphenylamino)-1-oxopropan-2-yl)-1-tosylpyrrolidine-2-carboxamide (9k).** Yield: 77%; a brown solid, mp 122–124˚C; R$_f$ = 0.44 (Hexane: EtOAc, 1:9, v/v).IR (CHCl$_3$) $v_{max}$: 1232, 1339 (S = O), 1667 (C = O), 3358 (N-H) cm$^{-1}$.$^1$H NMR (400 MHz, DMSO-d$_6$): $\delta$ = 1.34(d, $J$ = 6.4 Hz, 3H), 1.48 (s, 1H), 1.73–178 (m, 3H), 2.42 (s, 3H),3.13–3.18 (m, 1H), 3.40–3.44(m, 1H), 3.72 (s, 6H), 4.11–4.13(m, 1H),4.43 (t, $J$ = 6.5 Hz, 1H), 6.90(d, $J$ = 8.4 Hz, 1H), 7.13 (d, $J$ = 8.0 Hz, 1H), 7.32 (s, 1H), 7.45 (d, $J$ = 7.5 Hz, 2H), 7.77 (d, $J$ = 7.5 Hz, 2H), 8.26 (d, $J$ = 6.7 Hz, 1H), 9.71 (s, 1H). $^{13}$C NMR (100 MHz, DMSO-d$_6$): 18.7, 24.6, 31.1, 49.3, 49.6, 55.8, 56.2, 61.7, 104.8, 111.6, 112.6, 127.9, 130.3, 132.9, 134.2, 144.1, 145.4, 149.0, 170.8, 171.4 (C = O). ESI-HR-MS calculated for $C_{23}H_{29}N_3O_4S$ (M$^+$+H): 444.1957, found: 444.1949.

**$N$-(1-(3,5-dimethylphenylamino)-1-oxopropan-2-yl)-1-tosylpyrrolidine-2-carboxamide (9l).** Yield: 68%; a white solid, mp 122–124˚C; R$_f$ = 0.48 (Hexane: EtOAc, 3:7, v/v). IR (CHCl$_3$) $v_{max}$: 1217, 1344 (S = O), 1672 (C = O), 3373 (N-H) cm$^{-1}$.$^1$H NMR (400 MHz, DMSO-d$_6$): $\delta$ = 1.33(d, $J$ = 6.9 Hz, 3H), 1.47–1.49 (m, 1H), 1.70–180 (m, 3H), 2.23 (s, 6H), 3.41 (s, 3H), 3.12–3.18(m, 1H), 3.38–3.42(m, 1H), 4.13–4.15(m, 1H), 4.39–4.46 (m,1H),6.71 (s,1H), 7.23 (s,2H), 7.41–7.46 (m, 2H), 7.73–7.78 (m, 2H), 8.22 (d, $J$ = 7.2 Hz, 1H), 9.74 (s, 1H). $^{13}$C NMR (100 MHz, DMSO-d$_6$): 18.7, 21.5, 21.6, 24.6, 31.1, 49.4, 49.6, 61.5, 117.5, 125.4, 127.7, 127.9, 130.3, 134.4, 138.2, 139.1, 144.1, 171.2, 171.4 (C = O). ESI-HR-MS calculated for $C_{23}H_{29}N_3O_4S$ (M$^+$+H): 444.1957, found: 444.1949.

**$N$-(1-(5-methylthiazol-2-ylamino)-1-oxopropan-2-yl)-1-tosylpyrrolidine-2-carboxamide(9m).** Yield: 44%; a white solid, mp 234–236 $^o$C; R$_f$ = 0.47 (Hexane: EtOAc, 3:7, v/v). IR (CHCl$_3$) $v_{max}$: 1217, 1345 (S = O), 1678 (C = O), 3395 (N-H) cm$^{-1}$.$^1$H NMR (400 MHz, DMSO-d$_6$): $\delta$ = 1.33(d, $J$ = 7.1 Hz, 3H), 1.45–1.52 (m, 1H), 1.66–181 (m, 3H), 2.34(d, $J$ = 1.2 Hz, 3H), 2.41 (s, 3H), 3.11–3.17(m, 1H), 3.34–3.40 (m, 1H), 4.16 (dd, $J^1$ = 8.3 Hz; $J^2$ = 3.6 Hz, 1H), 4.50 (p, $J$ = 7.1 Hz, 1H), 7.14 (d, $J$ = 1.3 Hz, 1H), 7.44 (d, $J$ = 8.1 Hz, 2H), 7.75 (d, $J$ = 8.1 Hz, 2H), 8.30 (d, $J$ = 7.0 Hz, 1H), 11.95 (s, 1H). $^{13}$C NMR (100 MHz, DMSO-d$_6$): 11.5, 18.3, 21.5, 24.6, 31.1, 48.7, 49.5, 61.2, 126.9, 127.9, 130.3, 134.6, 135.2, 144.0, 156.4, 171.3, 171.6 (C = O). MS (ESI+): $m/z$ = 437.4. ESI-HR-MS calculated for $C_{19}H_{24}N_4O_4S_2$(M$^+$+H): 437.1317, found: 437.1314.

**$N$-(1-(benzo[d]thiazol-2-ylamino)-1-oxopropan-2-yl)-1-tosylpyrrolidine-2-carboxamide (9n).** Yield: 63%; a white solid, mp 110–112˚C; R$_f$ = 0.64 (Hexane: EtOAc, 3:7, v/v). IR (CHCl$_3$) $v_{max}$: 1267, 1344 (S = O), 1666 (C = O), 3293 (N-H) cm$^{-1}$.$^1$H NMR (400 MHz, DMSO-d$_6$): $\delta$ = 1.39(d, $J$ = 7.0 Hz, 3H), 1.48–1.55 (m, 1H), 1.69–183 (m, 4H), 2.42 (s, 3H), 3.12–3.18(m, 1H), 4.19 (dd, $J^1$ = 8.2 Hz; $J^2$ = 3.6 Hz, 1H), 4.57 (p, $J$ = 7.0 Hz, 1H), 7.30–7.34 (m, 1H), 7.43–7.43 (m,3H), 7.75–7.78 (m, 3H), 7.98 (d, $J$ = 7.4 Hz, 1H), 8.40 (d, $J$ = 6.8 Hz, 1H), 12.41 (s, 1H). $^{13}$C NMR (100 MHz, DMSO-d$_6$): 18.1, 21.5, 24.6, 31.1, 49.0, 49.5, 61.2, 121.1, 122.2, 124.1, 126.6, 127.9, 130.3, 131.9, 134.7, 144.0, 149.0, 171.7, 172.6 (C = O). MS (ESI+): $m/z$ = 473.3. ESI-HR-MS calculated for $C_{22}H_{24}N_4O_4S_2$(M$^+$+H): 473.1317, found: 473.1312.

**N-(1-(3-adamantan-1-yl)-1-oxopropan-2-yl)-1-tosylpyrrolidine-2-carboxamide (9o).** Yield: 54%; a white solid, mp 69–71˚C; R$_f$ = 0.43 (Hexane: EtOAc, 3:7, v/v). IR (CHCl$_3$) $v_{max}$: 1160, 1216, 1384 (S = O), 1664 (C = O), 3398 (N-H) cm$^{-1}$.$^1$H NMR (400 MHz, DMSO-d$_6$): $\delta$ = 0.84–0.87 (m, 1H), 1.21 (d, $J$ = 7.0 Hz, 3H), 1.45–1.49 (m, 1H), 1.62 (s, 6H), 1.72–1.79 (m, 2H),

1.93 (s, 6H), 2.01 (s, 3H), 2.42 2.01 (s, 3H), 3.10–3.16 (m, 1H), 3.37–3.45 (m, 1H), 4.07 (dd, $J^1$ = 8.2 Hz; $J^2$ = 3.7 Hz, 1H), 4.20–4.29 (m, 1H), 7.23 (s, 1H), 7.45 (d, $J$ = 8.0 Hz, 2H), 7.76 (d, $J$ = 8.3 Hz, 2H), 8.00 (d, $J$ = 8.0 Hz, 1H). $^{13}$C NMR (100 MHz, DMSO-d$_6$): 19.2, 21.5, 24.6, 29.3, 31.1, 31.2, 36.5, 41.4, 44.2, 48.8, 49.6, 50.2, 51.2, 61.9, 69.3, 128.0, 130.4, 134.1, 144.2, 171.0, 171.4 (C = O). MS (ESI+): $m/z$ = 474.5. ESI-HR-MS calculated for $C_{25}H_{35}N_3O_4S$ (M$^+$+H): 474.2427, found: 474.2430.

*N*-(1-(naphthalen-1-ylamino)-1-oxopropan-2-yl)-1-tosylpyrrolidine-2-carboxamide
**(9p).** Yield: 35%; a white solid, mp 168–170˚C; R$_f$ = 0.38 (Hexane: EtOAc, 3:7, v/v). IR (CHCl$_3$) $v_{max}$: 1216, 1346 (S = O), 1676 (C = O), 3401 (N-H) cm$^{-1}$. $^1$H NMR (400 MHz, DMSO-d$_6$): $\delta$ = 1.39–1.44 (m, 4H), 1.62–168 (m, 1H), 1.72–179 (m, 2H), 2.35 (s, 3H), 3.06–3.12(m, 1H), 3.30–3.41(m, 1H), 4.11 (dd, $J^1$ = 8.4Hz; $J^2$ = 3.6 Hz, 1H), 4.60 (p, $J$ = 7.2 Hz, 1H), 7.37–7.50 (m, 5H), 7.58 (d, $J$ = 7.0 Hz, 1H), 7.71 (t, $J$ = 9.0 Hz, 3H), 7.87–7.89 (m, 1H), 7.98–8.00 (m, 1H), 8.29 (d, $J$ = 7.3 Hz, 1H), 9.85 (s, 1H). $^{13}$C NMR (100 MHz, DMSO-d$_6$): 18.7, 21.5, 24.6, 31.1, 49.4, 49.6, 61.6, 122.4, 123.2, 126.0, 126.1, 126.4, 126.6, 127.9, 129.2, 128.4, 128.6, 130.3, 133.6, 134.2, 134.4, 144.1, 171.7, 172.2 (C = O). ESI-HR-MS calculated for $C_{25}H_{27}N_3O_4S$ (M$^+$+H): 466.1801, found: 466.1797.

*1*-(4-nitrophenylsulfonyl)-*N*-(1-oxo-1-(phenylamino)propan-2-yl)pyrrolidine-2-car**boxamide (10a).** Yield: 73%; a white solid, mp 156–158˚C; R$_f$ = 0.44 (Hexane: EtOAc, 3:7, v/v). IR (CHCl$_3$) $v_{max}$: 1245, 1351 (S = O), 1668 (C = O), 3368 (N-H) cm$^{-1}$. $^1$H NMR (400 MHz, DMSO-d$_6$): $\delta$ = 1.34 (d, $J$ = 7.0 Hz, 3H), 1.58–1.63 (m, 1H), 1.81–1.87 (m, 3H), 3.23–3.29 (m, 1H), 3.43–3.48 (m, 1H), 4.27 (dd, $J^1$ = 7.1 Hz; $J^2$ = 4.0 Hz, 1H), 4.43 (p, $J$ = 7.2 Hz, 1H), 7.05 (t, $J$ = 7.4 Hz, 1H), 7.31 (t, $J$ = 7.5 Hz, 2H), 7.59 (d, $J$ = 7.6 Hz, 2H), 8.12–8.14 (m, 2H), 8.36 (d, $J$ = 7.5 Hz, 1H), 8.42–8.44 (m, 2H), 9.93 (s, 1H). $^{13}$C NMR (100 MHz, DMSO-d$_6$): 18.7, 24.6, 31.3, 49.4, 49.6, 61.4, 119.6, 123.9, 125.0, 129.2, 129.3, 139.3, 143.2, 150.4, 171.1, 171.4 (C = O). MS (ESI+): $m/z$ = 447.4. ESI-HR-MS calculated for $C_{20}H_{22}N_4O_6S$ (M$^+$+H): 447.1338, found: 481.1335.

*1*-(4-nitrophenylsulfonyl)-N-(1-oxo-1-(p-tolylamino)propan-2-yl)pyrrolidine-2-car**boxamide(10b).** Yield: 78%; a yellow solid, mp 140–142˚C; R$_f$ = 0.46 (Hexane: EtOAc, 3:7, v/v). IR (CHCl$_3$) $v_{max}$: 1217, 1352 (S = O), 1672 (C = O), 3380 (N-H) cm$^{-1}$. $^1$H NMR (400 MHz, DMSO-d$_6$): $\delta$ = 1.33 (d, $J$ = 7.0 Hz, 3H), 1.58–1.63 (m, 1H), 1.81–1.87 (m, 3H), 2.24 (s, 3H), 3.23–3.27 (m, 1H), 3.43–3.48 (m, 1H), 4.24–4.27 (m,1H), 4.41 (p, $J$ = 7.2 Hz, 1H), 7.11 (d, $J$ = 8.3 Hz, 2H), 7.47 (d, $J$ = 8.74 Hz, 2H), 8.12–8.14 (m, 2H), 8.34 (d, $J$ = 7.4 Hz, 1H), 8.41–8.44 (m, 2H), 9.83 (s, 1H). $^{13}$C NMR (100 MHz, DMSO-d$_6$): 18.7, 20.9, 24.6, 31.3, 49.3, 49.6, 61.4, 119.6, 125.0, 129.4, 129.6, 132.8, 136.8, 143.2, 150.4, 171.1 (C = O). MS (ESI+): $m/z$ = 461.4. ESI-HR-MS calculated for $C_{21}H_{24}N_4O_6S$ (M$^+$+H): 461.1495, found: 461.1498.

*N*-(1-(4-methoxyphenylamino)-1-oxopropan-2-yl)-1-(4-nitrophenylsulfonyl)pyrroli**dine-2-carboxamide(10c).** Yield: 52%; a yellow solid, mp 131–133˚C; R$_f$ = 0.35 (Hexane: EtOAc, 3:7, v/v). IR (CHCl$_3$) $v_{max}$: 1217, 1352 (S = O), 1671 (C = O), 3380 (N-H) cm$^{-1}$. $^1$H NMR (400 MHz, DMSO-d$_6$): $\delta$ = 1.33 (d, $J$ = 7.1 Hz, 3H), 1.57–1.63 (m, 1H), 1.81–1.88 (m, 3H), 3.25–3.29 (m, 1H), 3.43–3.49 (m, 1H), 3.72 (s, 3H), 4.26 (dd, $J^1$ = 7.5 Hz; $J^2$ = 4.1 Hz, 1H), 4.40 (p, $J$ = 7.3 Hz, 1H), 6.86–6.90 (m, 2H), 7.48–7.52 (m, 2H), 8.11–8.15 (m, 2H), 8.33 (d, $J$ = 7.5 Hz, 1H), 8.41–8.44 (m, 2H), 9.77 (s, 1H). $^{13}$C NMR (100 MHz, DMSO-d$_6$): 18.8, 24.6, 31.3, 49.3, 49.6, 55.6, 61.5, 114.3, 121.2, 125.0, 129.4, 132.4, 143.2, 150.4, 155.8, 170.8, 171.1 (C = O). MS (ESI+): $m/z$ = 477.4. ESI-HR-MS calculated for $C_{21}H_{24}N_4O_7S$ (M$^+$+H): 477.1444, found: 477.1445.

*N*-(1-(4-isopropylphenylamino)-1-oxopropan-2-yl)-1-(4-nitrophenylsulfonyl)pyrroli**dine-2-carboxamide (10d).** Yield: 68%; a brown solid, mp 86–88˚C; R$_f$ = 0.58 (Hexane: EtOAc, 3:7, v/v). IR (CHCl$_3$) $v_{max}$: 1246, 1351 (S = O), 1660 (C = O), 3368 (N-H) cm$^{-1}$. $^1$H NMR (400 MHz, DMSO-d$_6$): $\delta$ = 1.17 (d, $J$ = 6.9 Hz, 6H), 1.33 (d, $J$ = 6.9 Hz, 3H), 1.58–1.63

(m, 1H), 1.80–1.87 (m, 3H), 2.80–2.87 (m, 1H), 3.23–3.29 (m, 1H), 3.44–4.49 (m, 1H), 4.26 (dd, $J^1$ = 7.3 Hz; $J^2$ = 4.0 Hz, 1H), 4.41 (p, $J$ = 7.2 Hz, 1H), 7.17 (d, $J$ = 8.5 Hz, 2H), 7.49 (d, $J$ = 8.6 Hz, 2H), 8.13 (d, $J$ = 9.0 Hz, 2H), 8.34 (d, $J$ = 7.4 Hz, 1H), 8.43 (d, $J$ = 8.7 Hz, 2H), 9.84 (s, 1H). $^{13}$C NMR (100 MHz, DMSO-d$_6$): 14.4, 18.7, 24.4, 24.6, 31.3, 49.3, 49.6, 61.4, 119.8, 125.0, 126.9, 137.0, 143.2, 143.9, 150.4, 171.1 (C = O). MS (ESI+): $m/z$ = 489.4. ESI-HR-MS calculated for $C_{23}H_{28}N_4O_6S$ (M$^+$+H): 489.1808, found: 489.1807.

*N*-(1-(4-chlorophenylamino)-1-oxopropan-2-yl)-1-(4-nitrophenylsulfonyl)pyrrolidine-2-carboxamide(10e). Yield: 78%; a white solid, mp 145–147 $^o$C; R$_f$ = 0.44 (Hexane: EtAAc, 3:7, v/v). IR (CHCl$_3$) $\nu_{max}$: 1216, 1352 (S = O), 1680 (C = O), 3380 (N-H) cm$^{-1}$.$^1$H NMR (400 MHz, DMSO-d$_6$): $\delta$ = 1.27 (d, $J$ = 7.0 Hz, 3H), 1.53 (t, $J$ = 5.1 Hz, 1H), 1.75–1.79 (m, 3H), 3.17–3.21 (m, 1H), 3.36–3.41 (m, 1H), 4.18–4.21 (m, 1H), 4.34 (p, $J$ = 7.0 Hz, 1H), 7.30 (d, $J$ = 8.8 Hz, 2H), 7.56 (d, $J$ = 8.7 Hz, 2H), 8.06 (d, $J$ = 8.7 Hz, 2H), 8.32 (d, $J$ = 7.3 Hz, 1H), 8.36 (d, $J$ = 8.7 Hz, 2H), 10.02 (s, 1H). $^{13}$C NMR (100 MHz, DMSO-d$_6$): 18.5, 24.6, 31.3, 49.4, 49.5, 61.3, 121.2, 125.0, 127.4, 129.1, 129.3, 131.5, 138.3, 143.3, 150.4, 171.2, 171.5 (C = O). ESI-HR-MS calculated for $C_{20}H_{21}ClN_4O_6S$ (M$^+$+H): 481.0949, found: 481.0953.

*N*-(1-(4-fluorophenylamino)-1-oxopropan-2-yl)-1-(4-nitrophenylsulfonyl)pyrrolidine-2-carboxamide (10f). Yield: 66%; a white solid, mp 156–158˚C; R$_f$ = 0.41 (Hexane: EtAAc, 3:7, v/v). IR (CHCl$_3$) $\nu_{max}$: 1216, 1352 (S = O), 1676 (C = O), 3379 (N-H) cm$^{-1}$.$^1$H NMR (400 MHz, DMSO-d$_6$): $\delta$ = 1.27 (d, $J$ = 7.1 Hz, 3H), 1.53 (t, $J$ = 5.1 Hz, 1H), 1.75–1.79 (m, 3H), 3.17–3.21 (m, 1H), 3.37–3.40 (m, 1H), 4.18–4.21 (m, 1H), 4.34 (p, $J$ = 7.1 Hz, 1H), 7.08 (t, $J$ = 8.8 Hz, 2H), 7.54 (q, $J$ = 5.1 Hz, 2H), 8.06 (d, $J$ = 8.8 Hz, 2H), 8.30 (d, $J$ = 7.3 Hz, 1H), 8.36 (d, $J$ = 8.6 Hz, 2H), 9.93 (s, 1H). $^{13}$C NMR (100 MHz, DMSO-d$_6$): 18.6, 24.6, 31.3, 49.4, 49.6, 61.4, 115.7, 115.9, 121.3, 121.4, 125.0, 129.3, 135.7, 143.2, 150.4, 157.3, 159.7, 171.1, 171.3 (C = O). MS (ESI+): $m/z$ = 465.3. ESI-HR-MS calculated for $C_{20}H_{22}FN_4O_6S$ (M$^+$+H): 465.1244, found: 465.1244.

1-(4-Nitrophenylsulfonyl)-N-(1-oxo-1-(3-(trifluoromethyl)phenylamino)propan-2-yl) pyrrolidine-2-carboxamide (10g). Yield: 67%; a white solid, mp 78–80˚C; R$_f$ = 0.60 (Hexane: EtAAc, 3:7, v/v). IR (CHCl$_3$) $\nu_{max}$: 1163, 1341 (S = O), 1667 (C = O), 3364 (N-H) cm$^{-1}$.$^1$H NMR (400 MHz, DMSO-d$_6$): $\delta$ = 1.37 (d, $J$ = 7.0 Hz, 3H), 1.57–1.66 (m, 1H), 1.81–1.91 (m, 3H), 3.25–3.30 (m, 1H), 3.43–3.48 (m, 1H), 4.27 (dd, $J^1$ = 7.6 Hz; $J^2$ = 3.9 Hz, 1H), 4.41 (p, $J$ = 7.1 Hz, 1H), 7.41 (d, $J$ = 7.7 Hz, 1H), 7.56 (t, $J$ = 8.1 Hz, 1H), 7.78 (d, $J$ = 8.3 Hz, 1H), 8.09–8.15 (m, 3H), 8.40–8.45 (m, 3H), 10.02 (s, 1H). $^{13}$C NMR (100 MHz, DMSO-d$_6$): 18.4, 24.6, 31.3, 49.6, 61.3, 115.6, 120.2, 123.2, 125.0, 125.9, 129.3, 129.8, 130.1, 130.5, 140.1, 143.3, 150.4, 171.2, 172.5 (C = O). MS (ESI+): $m/z$ = 515.3. ESI-HR-MS calculated for $C_{21}H_{21}F_3N_4O_6S$ (M$^+$+H): 515.1212, found: 515.1209.

*N*-(1-(4-chlorobenzylamino)-1-oxopropan-2-yl)-1-(4-nitrophenylsulfonyl)pyrrolidine-2-carboxamide (10h). Yield: 54%; a white solid, mp 127–129 $^o$C; R$_f$ = 0.30 (Hexane: EtAAc, 3:7, v/v). IR (CHCl$_3$) $\nu_{max}$: 1165, 1216, 1353 (S = O), 1671 (C = O), 3403 (N-H) cm$^{-1}$.$^1$H NMR (400 MHz, DMSO-d$_6$): $\delta$ = 1.27 (d, $J$ = 7.0 Hz, 3H), 1.57–1.60 (m, 1H), 1.78–1.87 (m, 3H), 3.22–3.26 (m, 1H), 3.43–3.47 (m, 1H), 4.20–4.32 (m, 4H), 7.25 (d, $J$ = 8.4 Hz, 2H), 7.36 (d, $J$ = 8.5 Hz, 2H), 8.11 (d, $J$ = 9.0 Hz, 2H), 8.25 (d, $J$ = 7.4 Hz, 1H), 8.37–8.43 (m, 3H). $^{13}$C NMR (100 MHz, DMSO-d$_6$): 18.7, 24.6, 31.2, 41.8, 48.8, 49.6, 61.5, 125.0, 128.7, 129.3, 131.8, 138.8, 143.2, 150.4, 171.0, 172.5 (C = O). MS (ESI+): $m/z$ = 495.3. ESI-HR-MS calculated for $C_{21}H_{23}ClN_4O_6S$ (M$^+$+H): 495.1105, found: 495.1100.

*N*-(1-(3,4-dichlorophenylamino)-1-oxopropan-2-yl)-1-(4-nitrophenylsulfonyl)pyrroli-dine-2-carboxamide(10i). Yield: 64%; a white solid, mp 120–122˚C; R$_f$ = 0.47 (Hexane: EtAAc, 3:7, v/v). IR (CHCl$_3$) $\nu_{max}$: 1308, 1352 (S = O), 1677 (C = O), 3369 (N-H) cm$^{-1}$.$^1$H NMR (400 MHz, DMSO-d$_6$): $\delta$ = 1.28(d, $J$ = 7.1 Hz, 3H), 1.52-152(m,1H), 1.73–1.84 (m, 3H), 3.17–3.23 (m, 1H), 3.36–3.41 (m, 1H), 4.20(dd, $J^1$ = 7.3 Hz; $J^2$ = 3.7 Hz, 1H), 4.31 (p, $J$ = 7.2

Hz, 1H), 7.42 (dd, $J^1$ = 8.8 Hz; $J^2$ = 2.3 Hz, 1H), 7.51 (d, $J$ = 8.9 Hz, 1H), 7.92 (d, $J$ = 2.4 Hz, 1H), 8.04–8.07 (m,2H), 8.33–8.37 (m,3H), 10.19 (s, 1H). $^{13}$C NMR (100 MHz, DMSO-d$_6$): 18.3, 24.6, 31.3, 49.5, 61.3, 119.7, 120.8, 125.0, 125.3, 129.3, 131.2, 131.5, 139.4, 143.3, 150.4, 171.2, 171.9 (C = O). MS (ESI+): $m/z$ = 515.2. ESI-HR-MS calculated for C$_{20}$H$_{21}$ClN$_4$O$_6$S (M$^+$+H): 515.0559, found: 481.0556.

**N-(1-(3-chloro-4-fluorophenylamino)-1-oxopropan-2-yl)-1-(4-nitrophenylsulfonyl) pyrrolidine-2-carboxamide (10j).** Yield: 57%; a white solid, mp 83–85˚C; R$_f$ = 0.55 (Hexane: EtOAc, 3:7, v/v). IR (CHCl$_3$) $\nu_{max}$: 1164, 1216, 1352 (S = O), 1677 (C = O), 3377 (N-H) cm$^{-1}$.$^1$H NMR (400 MHz, DMSO-d$_6$): $\delta$ = 1.23–1.16 (m, 1H), 1.35 (d, $J$ = 7.1 Hz, 3H), 1.59–1.62 (m, 1H), 1.82–1.88 (m, 3H), 4.04 (q, $J$ = 7.1 Hz, 1H), 4.26 (dd, $J^1$ = 7.5 Hz; $J^2$ = 3.6 Hz, 1H), 4.38 (p, $J$ = 7.2 Hz, 1H), 7.38 (d, $J$ = 9.1 Hz, 1H), 7.46–7.49 (m, 1H), 7.92 (dd, $J^1$ = 6.8 Hz; $J^2$ = 2.5 Hz, 1H), 8.13 (d, $J$ = 8.87 Hz, 2H), 8.39–7.44 (m, 3H), 10.18 (s, 1H). $^{13}$C NMR (100 MHz, DMSO-d$_6$): 18.4, 24.6, 31.3, 49.5, 61.3, 117.4, 117.6, 119.9, 120.0, 121.0, 125.0, 129.3, 136.5, 143.3, 150.4, 171.2, 171.7 (C = O). MS (ESI+): $m/z$ = 499.3. ESI-HR-MS calculated for C$_{20}$H$_{20}$ClFN$_4$O$_6$S (M$^+$+H): 499.0854, found: 499.0861.

***N*-(1-(3,4-dimethoxyphenylamino)-1-oxopropan-2-yl)-1-(4-nitrophenylsulfonyl)pyrro-lidine-2-carboxamide(10k).** Yield: 78%; a brown solid, mp 129–131˚C; R$_f$ = 0.43 (Hexane: EtOAc, 1:9, v/v). IR (CHCl$_3$) $\nu_{max}$: 1048, 1216 (S = O), 1674 (C = O), 3389 (N-H) cm$^{-1}$.$^1$H NMR (400 MHz, DMSO-d$_6$): $\delta$ = 1.33 (d, $J$ = 7.0 Hz, 3H), 1.58–1.63 (m,1H), 1.81–1.88 (m, 3H), 3.24–3.28 (m, 1H), 3.44–3.49 (m, 1H), 3.71 (d, $J$ = 3.1 Hz, 6H), 4.25 (dd, $J^1$ = 7.4 Hz; $J^2$ = 3.1 Hz, 1H), 4.39 (p, $J$ = 7.2 Hz, 1H), 6.89 (d, $J$ = 8.8 Hz, 1H), 7.11 (dd, $J^1$ = 8.7 Hz; $J^2$ = 2.4 Hz, 1H), 7.29 (d, $J$ = 2.3 Hz, 1H), 8.11–8.15 (m, 2H), 8.33 (d, $J$ = 7.5 Hz, 1H), 8.41–8.44 (m, 2H), 9.77 (s, 1H). $^{13}$C NMR (100 MHz, DMSO-d$_6$): 18.7, 21.6, 31.3, 49.3, 49.6, 55.8, 56.2, 61.5, 104.8, 111.6, 112.6, 125.0, 129.4, 132.9, 143.1, 145.4, 149.0, 150.4, 170.8, 171.0 (C = O). MS (ESI+): $m/z$ = 507.4. ESI-HR-MS calculated for C$_{22}$H$_{26}$N$_4$O$_8$S (M$^+$+H): 507.1550, found: 507.1552.

***N*-(1-(3,5-dimethylphenylamino)-1-oxopropan-2-yl)-1-(4-nitrophenylsulfonyl)pyrroli-dine-2-carboxamide(10l).** Yield: 78%; a yellow solid, mp 84–86˚C; R$_f$ = 0.58 (Hexane: EtOAc, 3:7, v/v). IR (CHCl$_3$) $\nu_{max}$: 1217, 1352 (S = O), 1670 (C = O), 3374 (N-H) cm$^{-1}$.$^1$H NMR (400 MHz, DMSO-d$_6$): $\delta$ = 1.32 (d, $J$ = 7.1 Hz, 3H), 1.57–1.65 (m,1H), 1.81–1.88 (m, 3H), 2.23 (s, 6H), 3.23–3.29 (m, 1H), 3.43–3.48 (m, 1H), 4.25–4.28 (m, 1H), 4.39 (p, $J$ = 7.2 Hz, 1H), 6.70 (s, 1H), 7.21 (s, 2H), 8.11–8.14 (m, 2H), 8.30 (d, $J$ = 7.3 Hz, 1H), 8.41–8.44 (m, 2H), 9.77 (s, 1H). $^{13}$C NMR (100 MHz, DMSO-d$_6$): 18.7, 21.6, 22.5, 24.6, 31.3, 49.4, 49.6, 61.4, 117.4, 125.0, 125.0, 125.4, 129.3, 138.2, 139.3, 143.2, 150.4, 171.1, 171.2 (C = O). MS (ESI+): $m/z$ = 475.4. ESI-HR-MS calculated for C$_{22}$H$_{26}$N$_4$O$_6$S (M$^+$+H): 475.1651, found: 475.1653.

***N*-(1-(5-methylthiazol-2-ylamino)-1-oxopropan-2-yl)-1-(4-nitrophenylsulfonyl)pyrroli-dine-2-carboxamide (10m).** Yield: 57%; a white solid, mp 232–234˚C; R$_f$ = 0.35 (Hexane: EtOAc, 3:7, v/v). IR (CHCl$_3$) $\nu_{max}$: 1216, 1352 (S = O), 1672 (C = O), 3401 (N-H) cm$^{-1}$.$^1$H NMR (400 MHz, DMSO-d$_6$): $\delta$ = 1.26 (d, $J$ = 7.0 Hz, 3H), 1.53–1.60 (m, 1H), 1.73–1.84 (m, 3H), 2.27 (d, $J$ = 1.0 Hz, 2H), 3.19–3.22 (m, 1H), 3.33–3.39 (m, 1H), 4.15 (dd, $J^1$ = 7.7 Hz; $J^2$ = 3.5 Hz, 1H), 4.39 (p, $J$ = 6.9 Hz, 1H), 7.07 (d, $J$ = 1.3 Hz, 1H), 8.03–8.06 (m, 2H), 8.33–8.37 (m, 3H), 11.91 (s, 1H). $^{13}$C NMR (100 MHz, DMSO-d$_6$): 11.5, 18.2, 24.6, 31.2, 48.7, 49.5, 61.2, 125.0, 126.9, 129.3, 135.3, 143.4, 150.4, 156.4, 171.2 (C = O). MS (ESI+): $m/z$ = 468.3. ESI-HR-MS calculated for C$_{18}$H$_{21}$N$_5$O$_6$S$_2$ (M$^+$+H): 468.1011, found: 468.1011.

***N*-(1-(benzo[d]thiazol-2-ylamino)-1-oxopropan-2-yl)-1-(4-nitrophenylsulfonyl)pyrroli-dine-2-carboxamide (10n).** Yield: 77%; a white solid, mp 144–146˚C; R$_f$ = 0.52 (Hexane: EtOAc, 3:7, v/v). IR (CHCl$_3$) $\nu_{max}$: 1267, 1352 (S = O), 1669 (C = O), 3360 (N-H) cm$^{-1}$.$^1$H NMR (400 MHz, DMSO-d$_6$): $\delta$ = 1.39 (d, $J$ = 7.2 Hz, 3H), 1.60–1.66 (m, 1H), 1.81–1.93 (m, 3H), 3.25–3.31 (m, 1H), 3.39–3.46 (m, 1H), 4.30 (dd, $J^1$ = 7.6 Hz; $J^2$ = 3.6 Hz, 1H), 4.52 (p, $J$ = 6.9 Hz, 1H), 7.29–7.33 (m, 1H), 7.42–7.47 (m, 1H), 7.76 (d, $J$ = 8.0 Hz, 1H), 7.98 (dd, $J^1$ =

7.9 Hz; $J^2$ = 0.5 Hz, 1H), 8.11 (t, $J$ = 2.0 Hz, 1H), 8.13 (t, $J$ = 2.4 Hz, 1H), 8.42 (t, $J$ = 2.0 Hz, 1H), 8.44 (t, $J$ = 2.4 Hz, 1H), 8.51 (d, $J$ = 6.7 Hz, 1H), 12.46 (s, 1H). $^{13}$C NMR (100 MHz, DMSO-$d_6$): 18.0, 24.6, 31.3, 49.0, 49.4, 61.1, 121.1, 122.2, 124.1, 125.0, 126.6, 129.3, 131.9, 143.5, 149.0, 150.4, 158.2, 171.3, 172.5 (C = O). ESI-HR-MS calculated for $C_{21}H_{21}N_5O_6S_2$ ($M^+$+H): 504.1011, found: 504.1006.

**N-(1-(3-adamantan-1-yl)-1-oxopropan-2-yl)-1-(4-nitrophenylsulfonyl) pyrrolidine-2-carboxamide (10o).** Yield: 50%; a white solid, mp 111–113 $^o$C; $R_f$ = 0.53 (Hexane: EtOAc, 3:7, v/v). IR (CHCl$_3$) $\nu_{max}$: 1165, 1216, 1310 (S = O), 1658 (C = O), 3397 (N-H) cm$^{-1}$.$^1$H NMR (400 MHz, DMSO-$d_6$): $\delta$ = 1.20 (d, $J$ = 6.8 Hz, 3H), 1.59 (d, $J$ = 12.4 Hz, 8H), 1.78–1.80 (m, 4H), 1.91 (s, 6H), 2.00 (s, 3H), 4.20–4.29 (m, 2H), 7.23 (s, 1H), 8.06 (d, $J$ = 7.8 Hz, 1H), 8.13 (d, $J$ = 8.6 Hz, 2H), 8.43 (d, $J$ = 8.8 Hz, 2H). $^{13}$C NMR (100 MHz, DMSO-$d_6$): 19.1, 24.6, 29.2, 31.2, 36.4, 41.4, 48.9, 49.6, 51.2, 61.8, 114.4, 125.1, 127.9, 129.4, 150.5, 170.8 (C = O). MS (ESI+): $m/z$ = 505.4. ESI-HR-MS calculated for $C_{24}H_{32}N_4O_6S$ ($M^+$+H): 505.2121, found: 505.2125.

***N*-(1-(naphthalen-1-ylamino)-1-oxopropan-2-yl)-1-(4-nitrophenylsulfonyl)pyrrolidine-2-carboxamide(10p).** Yield: 78%; a brown solid, mp 137–139˚C??; $R_f$ = 0.44 (Hexane: EtOAc, 3:7, v/v). IR (CHCl$_3$) $\nu_{max}$: 1216, 1352 (S = O), 1678 (C = O), 3380 (N-H) cm$^{-1}$.$^1$H NMR (400 MHz, DMSO-$d_6$): $\delta$ = 1.71 (d, $J$ = 7.0 Hz, 3H), 1.60–1.62 (m, 1H), 1.85–1.91 (m, 3H), 3.24–3.29 (m, 1H), 3.47–3.51 (m, 1H), 4.30 (t, $J$ = 5.7 Hz, 1H), 4.65 (p, $J$ = 3.1 Hz, 1H), 7.48–7.57 (m, 3H), 7.64 (d, $J$ = 7.2 Hz, 1H), 7.78 (d, $J$ = 8.1 Hz, 1H), 7.94 (t, $J$ = 5.2 Hz, 1H), 8.04 (t, $J$ = 4.2 Hz, 1H), 8.13 (d, $J$ = 8.7 Hz, 2H), 8.43 (d, $J$ = 8.8 Hz, 3H), 10.02 (s, 1H). $^{13}$C NMR (100 MHz, DMSO-$d_6$): 18.7, 24.6, 31.3, 49.3, 49.5, 61.4, 122.3, 123.2, 125.0, 126.0, 126.4, 126.5, 128.4, 128.6, 129.3, 133.6, 134.2, 143.2,150.4, 171.3, 172.1 (C = O). MS (ESI+): $m/z$ = 497.4. ESI-HR-MS calculated for $C_{24}H_{24}N_4O_6S$ ($M^+$+H): 497.1491, found: 497.1494.

## *In-vitro P. falciparum* assay

The sample concentration that inhibits the growth of chloroquine sensitive strains of *P. falciparum* development by 50% (IC$_{50}$) was measured and used to determine the antimalarial potencies of the new derivatives as thus: Sorbitol synchronized, 0.1% parasitemia, ring stage *P. falciparum* strain W2 parasites were cultured under the atmosphere of 3% O$_2$, 6% CO$_2$ and 91% N2 in RPMI-1640 medium supplemented with 10% human serum in the presence of inhibitors for 48 h without media change. Inhibitors were added from 1000 x DMSO stocks. After 48 h, the culture medium was removed and replaced with 1% formaldehyde in PBS pH 7.4 for an additional 48 h at room temperature to fix cells. Fixed parasites were transferred into 0.1% Triton-X-100 in PBS containing 1 nM YOYO-1 dye (Molecular Probes). Parasitemia was determined from dot plots (forward scatter vs. fluorescence) acquired on a FACS sort flow cytometer using Cell Quest software (Beckton Dickinson) [28].

## DPPH radical scavenging assay

The ability of the new dicarboxamides to reduce the 2,2-diphenyl-1-picrylhydrazyl (DPPH) radical was used to assess their antioxidant effect. Different test tubes containing solutions of each of the compounds in different concentrations (5, 10, 15, 20, and 25 μg/mL) were prepared in DMSO. 1 mL of freshly prepared DPPH solution (0.004% w/v) was added into each of these test tubes and shaken to mix properly. The test tubes were allowed to incubate in the dark room temperature for 30 mins. A blank solution containing everything else in the sample solution except the test compounds was also prepared and DPPH was used for the baseline correction. UV-Visible spectrometer (v. 6405 Jenway) was used to take record absorbance at 517 nm against the blank solution. Percentage inhibition (PI) of DPPH radical activity was used to

measure the radical scavenging activities of the dicarboxamides [29].

$$\text{DPPH radical scavenging activity (\%)} = [(A_{\text{DPPH}} - A_{\text{sample}})/A_{\text{DPPH}}] * 100$$

Where $A_{\text{DPPH}}$ = absorbance of control and $A_{\text{sample}}$ = absorbance of samples/ascorbic acid.

### Homology modeling

A query sequence of NMT from *P. falciparum* (access code: AAF18461.1) and template of X-ray crystal structure of NMT from *P. vivax* complexed with its cofactor (myristoyl Co-enzyme A) and an inhibitor (a pyrazole sulphonamide) (pdb code: 2ynd) [30] were employed to construct PfNMT homology model using modeler (v. 9.21) [31]. The template suitability was evaluated through the computed sequence-identity and -similarity. Twenty protein models were built and the best selected based on the evaluation by the Modeller objective function and Discrete Optimized Protein Energy profile. The best PfNMT homology model was energy minimized using Gromos force field 53a6 in Gromacs (v. 5.1, 2019) [32]. Ramanchandran plot was used to assess the quality of the model.

### Building 3-dimensional structures

The graphical user interface and MMFF94 forcefield packages in molecular operating environment (MOE: v. 2014.0901) software [33] were used to build the 3-dimensional chemical structures of the dicarboxamides and energy minimized them to an energy gradient of 0.001 kcal/mol while the QuSAR module in the same software was employed to compute the following molecular descriptors: lipophilicity (logP), molecular weight (MW), hydrogen bond acceptor/donor (HBA/HBD).

### Docking

Docking the new derivatives into PfNMT homology model was performed using AutoDock 4.2 program [34]. A grid box of 40 x 40 x 40 points with 0.375 Å point spacing placed in the center of mass center -25.835, 14.762, -25.529 such that it covered the peptide-substrate binding site. AutoGrid package was employed to calculate the potential grid maps for the interaction of ligand while AutoDock package was set to perform 250 hybrid GA-LS runs, making a maximum of 2.5 M energy evaluations and 27 000 generations. Maximum number of rotatable bonds was set at 6 while root mean square deviation tolerance for cluster grouping was set at 2Å. All other parameters were left at default setting.

## Results and discussion

### Synthesis and characterization of new dipeptide derivatives

Pyrrolidine moiety was introduced into the sulphonamide dicarboxamides because it occurs frequently in synthetic and isolated natural peptides that have antimalarial activity [35, 36]. The new sulphonamide pyrolidine carboxamide compounds were synthesised as presented in Fig 1. First, proline reacted with either toluene sulfonyl chloride or para-nitrophenyl sulfonyl chloride to produce either compounds **4** and **5** which were separately coupled with compound **8** via EDC.HCl/HOBT activation to obtain compounds **9a-p** and **10a-p** respectively. Intermediate **8** was obtained by reaction of boc-alanine with phenylamines followed by subsequent deprotection. The target molecules were fully characterised by combine spectroscopic and MS data and these are presented in S1 File. The IR absorption peaks for S = O, C = O and N-H functionalities were found in the range of 1200–1400 cm$^{-1}$, 1660–1680 cm$^{-1}$ and 3300–3400 cm$^{-1}$ respectively.

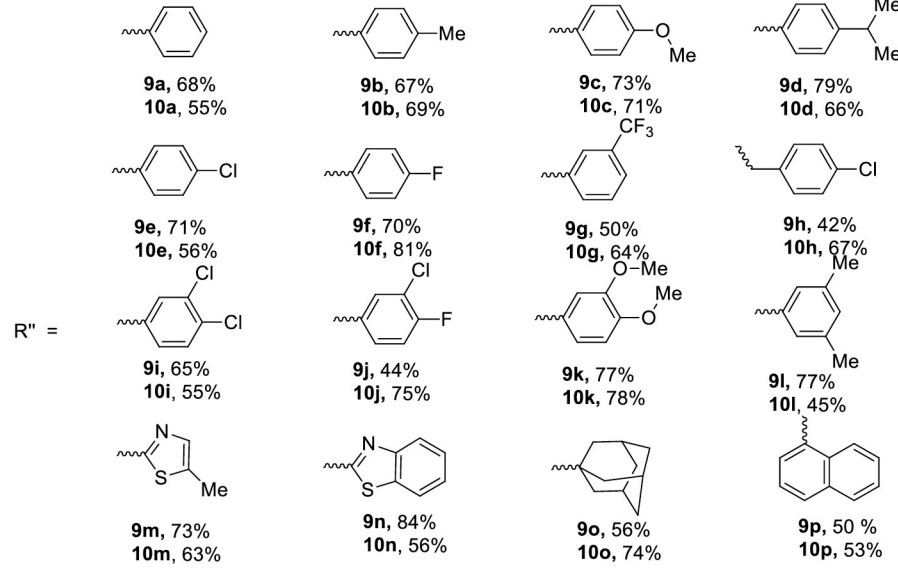

Scheme 1: Synthesis of toluenesulphonamidedipeptide, reagents and conditions - (a) $Na_2CO_3$, 0°C - r.t. , 6 h ; (b) EDC.HCl, HOBt, DIPEA, DCM, RT, 18 h (c) DCM/TFA (1:1 v/v) (d) EDC.HCl, HOBt, DIPEA, DCM, RT, 18 h

**Fig 1. Scheme 1.**

The methyl protons in toluenesulfonyl moiety integrated as singlet for three protons at about 2.40 ppm in the [1]H NMR of compounds **9a-p**. The methyl group that sandwich between the two amide groups appeared in the lowest field as doublet for all the compounds at around 1.35 ppm except for compound **9p**. The two ortho OMe substituent of the anilino ring of compounds **9k** and **10k** integrated as 6 protons singlets at 3.71 and 3.72 ppms respectively. The aromatic protons absorption peaks appeared within 7.00–8.30 ppm while the N-H protons showed up at the highest frequencies with chemical shifts in the range of 8.30–10.00 ppm in the proton NMR spectra. In addition, the [13]C NMR furnished the carbonyl carbons in all the synthesised compounds at the lowest field in all the spectra. Every other spectral data including MS are consistent with the molecular structures of the prepared compounds.

## Drug-likeness of the new sulphonamide-carboxamides

Since many drug candidates with very high therapeutic potential failed to succeed as drug due to poor pharmacokinetic profile it has become customary to evaluate the property early in drug discovery program to avoid wasting resources on wrong candidates. Hence, four molecular descriptors used to assess oral bioavailability of molecules according to Lipinski's rule of five (ro5) were computed for the new derivatives Table 1. The rule posits that molecules that have HBA $\leq$ 10, HBD $\leq$ 5, logP $\leq$ 5 and MW $\leq$ 500 Da or more than two of these criteria will

**Table 1. Basic physicochemical features of the new dicarboxamides.**

| Compound codes | HBA | HBD | logP(o/w) | MW (Da) |
|---|---|---|---|---|
| 9a | 7 | 2 | 1.97 | 415.51 |
| 9b | 7 | 2 | 2.27 | 429.54 |
| 9c | 8 | 2 | 1.92 | 445.54 |
| 9d | 7 | 2 | 3.11 | 457.59 |
| 9e | 7 | 2 | 2.56 | 449.95 |
| 9f | 7 | 2 | 2.12 | 433.5 |
| 9g | 7 | 2 | 2.9 | 483.51 |
| 9h | 7 | 2 | 2.69 | 463.98 |
| 9i | 7 | 2 | 3.19 | 484.4 |
| 9j | 7 | 2 | 2.75 | 467.94 |
| 9k | 9 | 2 | 1.67 | 475.56 |
| 9l | 7 | 2 | 2.6 | 443.56 |
| 9m | 8 | 2 | 1.13 | 436.55 |
| 9n | 8 | 2 | 2.6 | 472.59 |
| 9o | 7 | 2 | 2.74 | 473.63 |
| 9p | 7 | 2 | 3.19 | 465.57 |
| 10a | 10 | 2 | 1.61 | 446.48 |
| 10b | 10 | 2 | 1.9 | 460.51 |
| 10c | 11 | 2 | 1.56 | 476.51 |
| 10d | 10 | 2 | 2.75 | 488.56 |
| 10e | 10 | 2 | 2.2 | 480.92 |
| 10f | 10 | 2 | 1.76 | 464.47 |
| 10g | 10 | 2 | 2.54 | 514.48 |
| 10h | 10 | 2 | 2.33 | 494.95 |
| 10i | 10 | 2 | 2.82 | 515.37 |
| 10j | 11 | 2 | 0.76 | 467.52 |
| 10k | 12 | 2 | 1.3 | 506.53 |
| 10l | 10 | 2 | 2.24 | 474.53 |
| 10m | 7 | 2 | 2.1 | 429.54 |
| 10n | 11 | 2 | 2.23 | 503.56 |
| 10o | 10 | 2 | 2.37 | 504.6 |
| 10p | 10 | 2 | 2.83 | 496.54 |

likely be orally bioavailable [37]. All the newly synthesised compounds possess two HBD and logP in the range of 0.76–3.19. MW and HBA values also fall within the acceptable region except for five compounds with 503.56–515.37 Da and four compounds with 11–12 values respectively. However, following ro5 all the new compounds are likely druglike since none of them violated more than one of the criteria. Hence, they are worthy of further attention as drug candidates.

## Biological screening for antiplasmodial and antioxidant activities

Based on the fact that the pathophysiology mechanism through which plasmodia cause malaria results in cell oxidative stress, efforts are currently channelled to discovering agents with dual antiplasmodial and antioxidant activities [26, 27, 38]. The compounds were screened *in vitro* against chloroquine sensitive strains of *P. falciparum* malaria parasite at a maximum concentration of 20 μM and their $IC_{50}$ determined while anti-oxidant activity was evaluated by

checking for their ability to scavenge DPPH radical (Table 2). Although activity was not better than reference drug, chloroquine ($IC_{50}$ = 0.06 μM), as expected of compounds bearing sulphonamide and carboxamide functionalities, it is worthy to note that sixteen of the sulphonamide-carboxamides killed the pathogen at single-digit values of half-maxima inhibitory concentration in micromolar ($IC_{50}$ = 2.40–8.30 μM), nine showed $IC_{50}$ between 10 and 20 μM concentration while only seven have $IC_{50}$ > 20 μM and so were reported as having no activity (na). The close range of $IC_{50}$s suggests there is no outstanding structure-activity relationship, however, it was observed that attaching thiazol (**9m**, **10m**, **9n**, and **10n**) and adamantanyl (**9o** and **10o**) substituent moieties at the N-terminal position of the parent-structure relatively led to an

**Table 2. *In vitro* antiplasmodial and antioxidant activities of compounds against chloroquine sensitive 3D7 strain of *Plasmodium falciparum*.**

| Compound codes | Anti-P. falciparum activity $IC_{50}$ (μM) | Antioxidant activity $IC_{50}$ of DPPH (μg/mL) |
|---|---|---|
| **9a** | 16.4 | 12.51 |
| **9b** | 12.6 | 12.02 |
| **9c** | 16.8 | 12.95 |
| **9d** | 12.2 | 12.37 |
| **9e** | 6.2 | 12.59 |
| **9f** | 8.6 | 14.91 |
| **9g** | 6.6 | 12.43 |
| **9h** | 11.6 | 13.16 |
| **9i** | 6.4 | 14.95 |
| **9j** | 6.8 | 12.85 |
| **9k** | na | 11.7 |
| **9l** | na | 13.71 |
| **9m** | 3.2 | 12.34 |
| **9n** | 2.8 | 13.43 |
| **9o** | 5 | 11.75 |
| **9p** | na | 13.22 |
| **10a** | 10.6 | 12.71 |
| **10b** | na | 6.48 |
| **10c** | 18 | 8.49 |
| **10d** | 11 | 3.02 |
| **10e** | 6.4 | 12.36 |
| **10f** | 6.6 | 12.16 |
| **10g** | 4.4 | 12.71 |
| **10h** | 15 | 12.65 |
| **10i** | 5 | 12.75 |
| **10j** | 3.2 | 6.44 |
| **10k** | na | 12.77 |
| **10l** | na | 12.52 |
| **10m** | 2.8 | 12.28 |
| **10n** | 2.4 | 12.7 |
| **10o** | 3.6 | 4.32 |
| **10p** | na | 12.42 |
| Chloroquine phosphate | 0.06 | - |
| Ascorbic acid | - | 1.06 |

na = no activity

increased antiplasmodial activity. Similarly, para-nitrophenyl derivatives in general were found to exhibit higher activity than the ones bearing toluenesuphonamide. For example, **10m**, **10n** and **10o** with para-nitrophenylsulphonamide attachment respectively have higher inhibitory potencies (IC$_{50}$ = 2.80, 2.40 and 3.6 μM) than **9m**, **9n** and **9o** (IC$_{50}$ = 3.20, 2.80 and 5.00 μM). On the other hand, compounds **10b**, **10c**, **10d**, **10j** and **10o** scavenged DPPH radicals at IC$_{50}$s (6.48, 8.49, 3.02, 6.44 and 4.32 μg/mL respectively) comparable to ascorbic acid (1.06 μg/mL) while all the compounds reduced more than 50% oxidative property of the tested radicals. It appears compound **10j** and **10o** are most suitable for our interest since only the two possess both antiplasmodial and antioxidant activities at relatively low micromolar and microgram concentration respectively.

## Homology modelling

The PfNMT homology model was constructed using modeller 9.21v with a PfNMT query sequence (accession code: AAF18461.1) and template X-ray crystal structure of NMT from *P. vivax* (PvNMT) in complex with an inhibitor, a pyrazole sulphonamide (accession code: 2ynd). There were sequence-identity and -similarity of 82% and 93% between the target and template which indicate the template sequence is suitable for the process. Moreover, there was only 0.350 Å root mean square deviation upon alignment of the homology modelled PfNMT and the template structure. In addition, the model possesses good stereochemical quality because more than 90% of its residues lie in the recommended region according to the Ramanchandran plot in Fig 2.

## Docking results

To check whether the new sulphonamide-carboxamide derivatives can bind to PfNMT, the default forcefield in AutoDock4.2v was used to dock and score compounds **9a** – **9p** and **10a** –

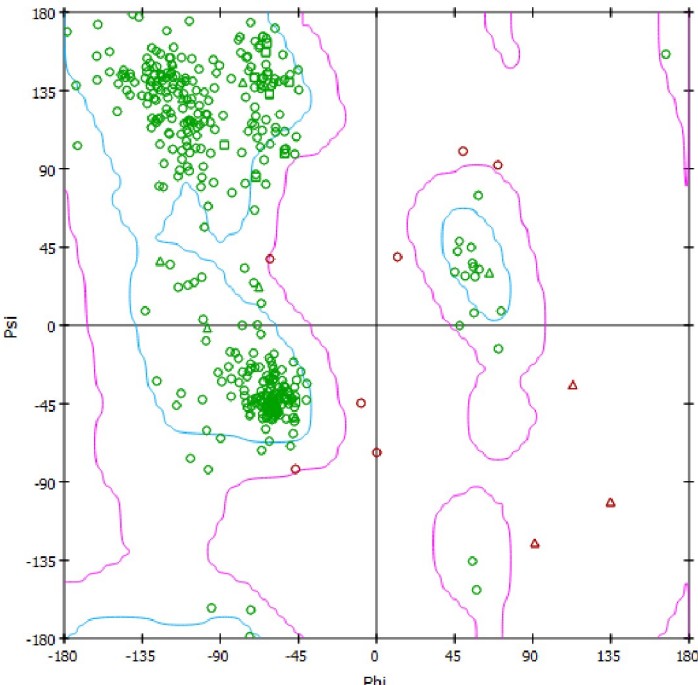

**Fig 2. The Ramanchandran plot of PfNMT homology model.**

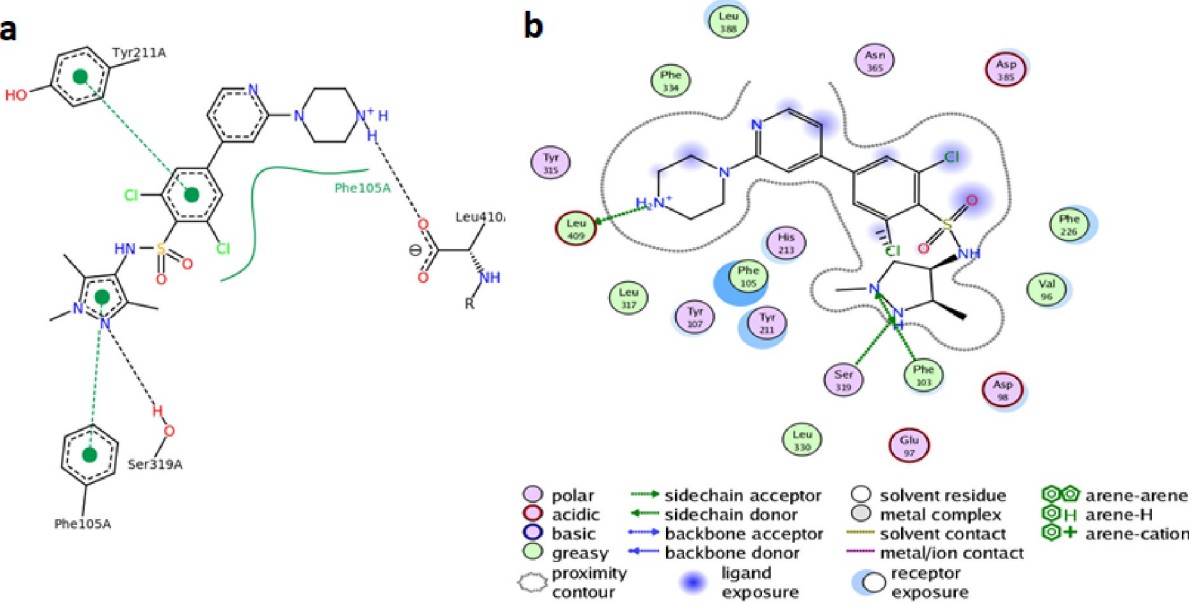

**Fig 3.** The reference ligand i.e. the pyrazole sulphonamide (a) X-ray crystallography predicted binding pose in P. vivax NMT and (b) docked pose predicted by AutoDock in the PfNMT hoology model.

**10p** toward the peptide-substrate binding site of the modelled protein. The co-crystallized ligand from PvNMT was used as reference in evaluating the performance of the dock-protocol. The retained dock protocol was through visual inspection based on the ability of the software to return ligand dock-pose comparable to X-ray crystallized pose within the rmsd tolerance of 2 Å for cluster grouping (Fig 3). The least theoretical binding energies of the derivatives ($\Delta G$ = -6.98 to -9.60 kcal/mol) retrieved from the highest populated clusters docking poses show they preferentially interacted with the protein (Table 3) and therefore are potential PfNMT binders. Though none of them has docking score as high as the reference ligand ($\Delta G$ = -10.91 kcal/mol, $K_i$ = 10.11 nM, ligand efficiency = 0.33), their low computed inhibition constant ($K_i$ = 0.09–1.18 μM), reasonable ligand efficiencies (0.23–0.28 kcal/mol) and interesting drug-like character make the new compounds a promising series deserving further investigation. Moreover, notice that compound **10o** which had earlier shown interesting superior dual biological activities, also emerged as topscorer in molecular docking study thereby highlighting the importance of this derivative.

## Conclusion

It is increasing becoming necessary for antimalaria agents to possess antioxidant property as well because of the oxidative stress which accompanies malaria pathophysiological mechanism and the search for such agents within compounds bearing both sulphonamide and carboxamide moieties is common due to proven pharmacological efficiencies of both functionalities. In this study, we presented the synthesis and full characterization of 32 new druglike sulphonamide-carboxamide derivatives and explored their antimalaria and antioxidant properties. The results show that some of the derivatives are able to kill *P. falciparum* at single-digit micromolar concentration and scavenge DPPH radicals at comparable degree to ascorbic acid. In addition, through molecular docking and scoring, we calculated the theoretical binding affinities of the compounds for PfNMT homology model and observed that 15 of them favourably bound to the target at submicromolar $K_i$ values. Particularly compound **10o**, with relatively significant

**Table 3. Docking results for the new toluenesulphonamide dicarboxamides.**

| Compund codes | ΔG (kcal/mol) | $K_i$ (µM) | Ligand Efficiency |
|---|---|---|---|
| 9a | -7.89 | 1.64 | 0.27 |
| 9b | -8.09 | 1.18 | 0.27 |
| 9c | -8.31 | 0.81 | 0.27 |
| 9d | -9.05 | 0.23 | 0.28 |
| 9e | -7.99 | 1.38 | 0.27 |
| 9f | -6.98 | 7.67 | 0.23 |
| 9g | -8.55 | 0.54 | 0.26 |
| 9h | -7.43 | 3.6 | 0.24 |
| 9i | -8.41 | 0.68 | 0.27 |
| 9j | -7.94 | 1.53 | 0.26 |
| 9k | -8.43 | 0.66 | 0.26 |
| 9l | -8.33 | 0.77 | 0.27 |
| 9m | -8.09 | 1.18 | 0.28 |
| 9n | -8.58 | 0.51 | 0.27 |
| 9o | -8.61 | 0.48 | 0.26 |
| 9p | -7.78 | 1.99 | 0.24 |
| 10a | -7.52 | 3.05 | 0.24 |
| 10b | -7.45 | 3.48 | 0.23 |
| 10c | -7.68 | 2.36 | 0.23 |
| 10d | -8.78 | 0.36 | 0.26 |
| 10e | -7.89 | 1.65 | 0.25 |
| 10f | -7.23 | 5.05 | 0.23 |
| 10g | -8.19 | 0.98 | 0.23 |
| 10h | -7.6 | 2.7 | 0.23 |
| 10i | -8.47 | 0.62 | 0.26 |
| 10j | -7.58 | 2.8 | 0.24 |
| 10k | -8.21 | 0.96 | 0.23 |
| 10l | -8.22 | 0.94 | 0.25 |
| 10m | -7.67 | 2.4 | 0.26 |
| 10n | -8.26 | 0.87 | 0.24 |
| 10o | -9.6 | 0.09 | 0.27 |
| 10p | -9.17 | 0.18 | 0.26 |
| rl | -10.91 | 0.01 | 0.33 |

NB: rl means reference ligand i.e. pyrazole-sulphonamide cocrystallised with PvNMT (pdb code: 2ynd)

dual pharmacological effect and topscorer towards PfNMT homology model, will be focused in our further studies to optimize activity. We hope that the results of this study will facilitate effort to design and discover new antimalaria candidate with antioxidant activity from carboxamides bearing sulphonamide moiety.

## Supporting information

**S1 File. Detail spectral data used to characterise and describe the new sulphonamide pyrolidine carboxamide.**
(DOCX)

## Author Contributions

**Conceptualization:** Efeturi A. Onoabedje, Uchechukwu C. Okoro.

**Data curation:** Efeturi A. Onoabedje, Akachukwu Ibezim.

**Formal analysis:** Efeturi A. Onoabedje, Akachukwu Ibezim.

**Funding acquisition:** Efeturi A. Onoabedje, Akachukwu Ibezim.

**Investigation:** Efeturi A. Onoabedje, Akachukwu Ibezim.

**Resources:** Sanjay Batra.

**Software:** Akachukwu Ibezim, Sanjay Batra.

**Supervision:** Uchechukwu C. Okoro.

**Validation:** Uchechukwu C. Okoro, Sanjay Batra.

**Writing – original draft:** Efeturi A. Onoabedje, Akachukwu Ibezim.

**Writing – review & editing:** Uchechukwu C. Okoro, Sanjay Batra.

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
