## [Decision Letter · Decision Letter 0]

11 Jun 2020

PONE-D-20-14557

New sulphonamide pyrolidine carboxamide derivatives: synthesis, molecular docking, antiplasmodial and antioxidant activities

PLOS ONE

Dear Dr. Ibezim,

Thank you for submitting your manuscript to PLOS ONE. After careful consideration, we feel that it has merit but does not fully meet PLOS ONE’s publication criteria as it currently stands. Therefore, we invite you to submit a revised version of the manuscript that addresses the points raised during the review process.

We look forward to receiving your revised manuscript.

Kind regards,

Mohammad Shahid, Ph.D.

Academic Editor

PLOS ONE

Journal Requirements:

2. Please include further information such as vendor, manufacturer and model details of all equipment and materials used in the study.

3. Please amend your Financial disclosure statement to only declare sources of funding for the study, and remove any other text.

4. We suggest you thoroughly copyedit your manuscript for language usage, spelling, and grammar. If you do not know anyone who can help you do this, you may wish to consider employing a professional scientific editing service.  

Reviewers' comments:

Reviewer's Responses to Questions

**Comments to the Author**

1. Is the manuscript technically sound, and do the data support the conclusions?

Reviewer #1: Partly

2. Has the statistical analysis been performed appropriately and rigorously? 

Reviewer #1: N/A

3. Have the authors made all data underlying the findings in their manuscript fully available?

Reviewer #1: Yes

4. Is the manuscript presented in an intelligible fashion and written in standard English?

Reviewer #1: Yes

5. Review Comments to the Author

Reviewer #1: The authors described the synthesis and biological evaluation of 32 dipeptides Ar1SO2-Pro-Ala-NHAr2. The synthesis is modular with moderate to good chemical yields. Some of these compounds exhibit single-digit micromolar IC50s against chloroquine sensitive strains of P. falciparum and act as radical scavengers well. Some computational studies were performed on the selected dipeptides.

One of the major concerns is that all the tested compounds are not very active against the malarial parasites or as antioxidant. It is hardly to call the functionalized dipeptide scaffold novel unless the authors can demonstrate they have good activity profiles. The screening of a 32-compound library with fixed amino acid backbone (Pro-Ala) is not diverse enough for SAR, and the expansion to a more inclusive compound set seems necessary to discover a more potent inhibitor. It is not entirely clear why the amino acid should be limited to Pro and Ala, at least some other amino acids should be considered. The toxicity of these test compounds should be evaluated.

The discussion in the Synthesis and characterization of new dipeptide derivatives (last paragraph under scheme 2) seems not delivering important information. It is focused on NMR data of the compounds. For example “Significantly, the methyl protons in toluenesulfonyl moiety integrated as singlet for three protons at about 2.40 ppm in the 1H NMR of compounds 9a-p and 10a-p”, why this information is significant? Do the authors try to confirm that the compounds are diastereomerically pure? And compounds 10a-p don’t even have a toluenesulfonyl moiety! The authors also tried to convey that a OMe is more deshielded than a Me on the aromatic ring, this is not necessary. In addition, the drawings in Scheme 2 need further improvement. First, as a medicinal chemistry manuscript, the identity of the test compounds is very important. No stereochemistry of the amino acid is labelled. The authors should clearly specify the configuration of the compounds since both enantiomers of the amino acids are available and different stereoisomers can alter the activity significantly. In structure 4/5, the X is missing. For the final products, the label should be changed as “9: X= Me, 10: X=NO2”. The current label is very confusing, since R’’=9 :x=Me is not correct.

Lipinski’s rule of five is an empirical guideline to assess the drug-likeness of a compound. However, there are many exceptions on both ways. Just using this guideline to evaluate the compounds conclude their drug likeness is not sufficient. One obvious liability is the instability of these peptide drugs in general: they are susceptible to enzymatic degradation. The authors should have a microsomal stability test on them.

6. PLOS authors have the option to publish the peer review history of their article (what does this mean?). If published, this will include your full peer review and any attached files.

Reviewer #1: No

---

## [Author Response · Author response to Decision Letter 0]

31 Oct 2020

We have addressed the concerns raised by both the editor and reviewers and have also attached relevant files appropriately.

---

## [Editor Report · Decision Letter 1]

19 Nov 2020

New sulphonamide pyrolidine carboxamide derivatives: synthesis, molecular docking, antiplasmodial and antioxidant activities

PONE-D-20-14557R1

Dear Dr. Ibezim,

We’re pleased to inform you that your manuscript has been judged scientifically suitable for publication and will be formally accepted for publication once it meets all outstanding technical requirements.

Kind regards,

Mohammad Shahid, Ph.D.

Academic Editor

PLOS ONE
---

## [Editor Report · Acceptance letter]

3 Dec 2020

PONE-D-20-14557R1 

New sulphonamide pyrolidine carboxamide derivatives: synthesis, molecular docking, antiplasmodial and antioxidant activities 

Dear Dr. Ibezim:

I'm pleased to inform you that your manuscript has been deemed suitable for publication in PLOS ONE. Congratulations! Your manuscript is now with our production department. 

Kind regards, 

on behalf of

Dr. Mohammad Shahid 

Academic Editor

PLOS ONE